# Reconstruction and coil combination of undersampled concentric-ring MRSI data using a Graph U-Net

**Paul Weiser** [1]                                    PAUL.WEISER@MEDUNIWIEN.AC.AT

**Stanislav Motyka** [2]                               STANISLAV.MOTYKA@MEDUNIWIEN.AC.AT

**Wolgang Bogner** [2]                                 WOLFGANG.BOGNER@MEDUNIWIEN.AC.AT

**Georg Langs** [1]                                    GEORG.LANGS@MEDUNIWIEN.AC.AT

[1] *Computational Imaging Research Lab - Department of Biomedical Imaging and Image-guided Therapy, Medical University of Vienna, Austria*

[2] *High Field MR Center - Department of Biomedical Imaging and Image-Guided Therapy, Medical University of Vienna, Austria*

**Editors:** Under Review for MIDL 2021

## Abstract

Geometric deep learning has recently gained influence, as it allows the extension of convolutional neural networks to non euclidean domains. In this paper graph neural networks (GNNs) are used for the image reconstruction and coil combination of undersampled concentric-ring k-space MRSI data. We show that graph U-nets perform better on undersampled data than GNNs. Specifically, results suggest that the omission of self-connecting edges results in a more stable behavior and better training for graph U-nets.

**Keywords:** MR Spectroscopy, Geometric Deep Learning, U-Net.

## 1. Introduction

MR spectroscopy imaging (MRSI) is an imaging modality that has many applications in medicine, as it allows the identification of various biochemical substances in vivo (Maudsley et al., 2009). In recent years, fueled by advances in deep learning (DL), new imaging techniques have been developed in medicine and also MRI has gained from this development (Lundervold and Lundervold, 2019). However, in MRSI irregular sampling schemes can be beneficial, and for those, DL based reconstruction is lacking. Here, we investigate geometrical deep learning for k-space reconstruction of undersampled concentric-ring sampled MRSI data. The current state of the art approach for reconstruction of undersampled multi-coil data is parallel imaging (with schemes such as *SENSE* and *GRAPPA*) (Uecker et al., 2014). The latter is a kernel based approach, and therefore naturally gives rise to deep learning based methods. In this paper, graph U-nets are proposed for the reconstruction of non-Cartesian k-space data.

## 2. Data and Method

Non-water surppressed MRSI data was collected from seven volunteers in ten random positions. The data of the first six volunteers was used for training and the data of the last

volunteer was split up into validation and testing set. In each scan concentric ring trajectories where used, in total 16 rings with 388 points per ring each were acquired. Graphs were defined by connecting point pairs with a distance less than 1.5 times Nyqusit criterion. These rings were undersampled, by fully sampling the inner 6 and then skipping every second of the outer rings. The output data for training and evaluation was computed by transforming the fully sampled data to image space, where *ESPIRiT* (Uecker et al., 2014) sensitivity maps were applied, and then back to k-space. We evaluated two models. The first network (referred to as *GNN*) consists of four gaussian mixture model (GMM) convolutional layers (Monti et al., 2017), each followed by a *tanh* activation function. The second model (*U-Net*) is a U-net (Ronneberger et al., 2015). Here five GMM convolutional layers are used, each followed by max-pooling or up-sampling with a window size of $4 \times 2$ and tanh or ReLU activation.

Additionally, a naive GRAPPA approach was implemented. Therefore, the circles were split up in segments with a width of 24 nodes and a length of all 16 rings. In the fully sampled area a kernel with six weights per coil was optimized and used to reconstruct the missing rings of its segment.

## 3. Evaluation and Results

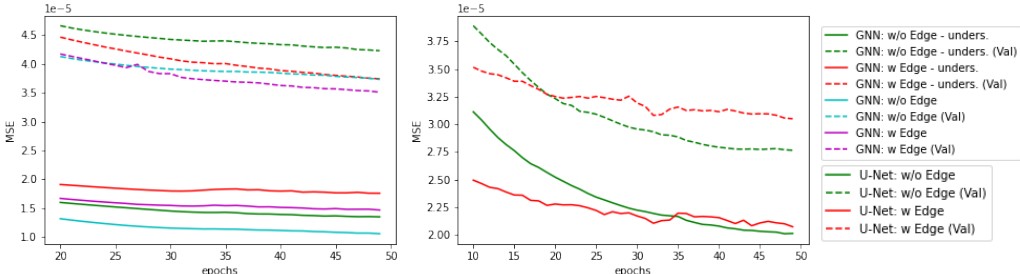

Figure 1: Training/Validation loss of GNN and U-net.

First the four layer GNN was trained on fully sampled and undersampled data, as well as with and without self-connecting edges. The training and validation loss, computed by the mean squared difference, is shown in figure 3 on the left. Self-connecting edges improve the validation loss during training of the network in both cases.

In figure 3 on the right, the training- and validation loss of the graph U-net with undersampled data with and without self-connections is plotted. In this case, the omitted self-connecting edge leads to a reduced and more stable loss.

On the test set we reconstructed images from understampled data by the naive GRAPPA algorithm, the GNN with self-connecting edges and the graph U-net without self-connecting edges and used

|  | **GRAPPA** | **GNN** | **U-net** |
|---|---|---|---|
| Position 1 | 5908.9 | 137.0 | 67.9 |
| Position 2 | 6713.1 | 55.7 | 53.0 |
| Position 3 | 2610.4 | 42.5 | 27.8 |
| Position 4 | 7548.6 | 126.9 | 39.1 |

Table 1: Mean squared error of four scanned head positions in the test set.

Fourier transform in all spacial dimensions to reconstruct the image. The mean squared error of each scanned position is presented in table 3 and shows that the U-net performs best. In figure 3 qualitative results of each approach are compared.

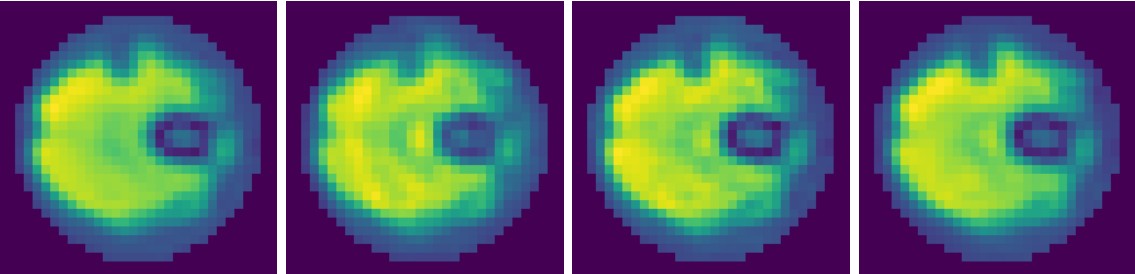

Figure 2: Self-normalized images. Left to right: Ground Truth, GRAPPA, GNN, U-net.

## 4. Discussion

Compared to the naive GRAPPA approach and the GNN, the graph U-net leads to an improvement of the reconstruction of undersampled concentric-ring MRSI, due to its ability to identify high-level features.

The omission of self-connecting edges leads to a decreased and more stable loss with the U-net. This may be the case, because the network is forced to search for informative features in the neighborhood of each node, instead of simply passing on information.

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
