# OpenReview forum: "Reconstruction and coil combination of undersampled concentric-ring MRSI data using a Graph U-Net"
_MIDL.io/2021/Conference/Short — MIDL 2021 Poster_

### Official Review · Reviewer_CCy6 · 2021-04-29

**Confidence:** 4
**Final Rating:** 3

**Summary:**

The submitted manuscript"Reconstruction and coil combination of undersampled concentric-ring MRSI data using a Graph U-Net" combines timely methods to outperform established reconstruction methods of parallel imaging and boost application of challenging MRSI. The method is described briefly and data describtion appears rather short. The presented results after evaluation show clear benefits, but should be interpreted carefully as some important information are missing and genralization is not guaranteed. This could have been pointed out in the discussion statement.

**Strengths:**

The manuscript sets a clear focus on the reconstruciton of undersampled spectroscopic imaging data in MR.  The problem is clearly motivated. The choosen approach to employ a neural network and graph data approach is interesting and promissing. The presentend results underpin this impression.

**Weaknesses:**

The submitted manuscript has multiple shortcomings foremost in the detailed describtion of the "Data and Methods" section. This may be due to word or page limitations, but appears nontheless limiting for interested reader.
In the Data and Methods section little information about the experimental procedure (imaging parameters, data formats, etc.) are given. However, this would be of interest for the readers to redo or learn from the presented work.
The authros claim (Tabel 1) that a quantitaitve comparison of the presented methods to established/conventional methods reveals better performance (MSE). However, as no details are given I suscpect that appropriate (re)scaling, i.e. normalization of image data (or input data) was done, such that absolute values have limited value of interpretation. The qualitaitve results back the aforementioned quantitative results, but are no interpreted in detail or highlighted (i.e. by difference-to-reference maps).
The discussion appears not very balanced and conclusions (high-level features) are very general statements with limited value for interpretation.

**Deanonymize Review:**

yes

**Justification Of The Rating:**

I weakly accept the submitted manuscript as I see a focussed and somehow novel approach to a relevant problem in MRSI. The manuscript has some shortcomings which may also be related to word/page limits, but these also prevent potential reproducibility. Nontheless, I think this work is of interest for the community and the authors may answer open points and details during MIDL.

**Paper Type:**

methodological development

**Special Issue:**

no

---

### Meta-Review · Area_Chair_t9eK · 2021-05-09

**Recommendation:** Accept (Poster)
**Confidence:** 5

**Metareview:**

Unfortunately only one review could be obtained in time. However, I agree with the reviewer that the paper describes an interesting approach and fits well within MIDL.

---

### Decision · Program_Chairs · 2021-05-11

Accept (Poster)